# Selection for growth drives the emergence of genetic heredity in protocells

**Raquel Nunes Palmeira[1], Marco Colnaghi[1,2], Andrew Pomiankowski[1]\*, Nick Lane[1]**

**1** Centre for Life's Origins and Evolution, Department of Genetics, Evolution and Environment, Division of Biosciences, University College London, London, United Kingdom, **2** Department of Applied and Experimental Psychology, Vrije Universiteit Amsterdam, Amsterdam, The Netherlands

\* a.pomiankowski@ucl.ac.uk

## Abstract

Most theoretical work on the origin of heredity has focused on how genetic information can be maintained without mutational degradation in the absence of error-proofing systems. A simple and parsimonious solution assumes the first gene sequences evolved inside dividing protocells, which enables selection for functional sets. But this model of information maintenance does not consider how protocells acquired their genetic information in the first place. Clues to this transition are suggested by patterns in the genetic code, which indicate a strong link to autotrophic metabolism, with early translation based on direct physical interactions between amino acids and short RNA polymers, grounded in their hydrophobicity. Here, we develop a mathematical model to investigate how random RNA polymers inside autotrophically growing protocells could evolve better coding sequences for discrete functions. The model tracks a population of protocells that evolve towards two essential functions: $CO_2$ fixation (which drives monomer synthesis and cell growth) and copying (which amplifies replication and translation of sequences inside protocells). The model shows that distinct coding sequences can emerge from random RNA sequences driving increased protocell division. The analysis reveals an important restriction: growth-supporting functions such as $CO_2$ fixation must be more easily attained than informational processes such as RNA copying and translation. This uncovers a fundamental constraint on the emergence of genetic heredity: growth precedes information at the origin of life.

## Introduction

The emergence of genetic heredity was a critical step in the origin of life. Before the advent of genes, proto-living systems could only be a product of whatever chemistry was kinetically and thermodynamically favored, even if part of a complex reaction network. Once genes existed, natural selection could begin to shape life. The full genetic apparatus must have been present in the last universal common ancestor

**Data availability statement:** Code used in this paper is available at: https://github.com/raquel-npalmeira/first_growth_then_information and archived on Zenodo: https://doi.org/10.5281/zenodo.18940155.

**Funding:** RNP, MC, AP, and NL are supported by funding from the Biotechnology and Biological Sciences Research Council (BB/V003542/1), https://www.ukri.org/councils/bbsrc/. This funding includes ongoing salary support for RNP and previously provided salary support for MC. AP is also supported by funding from the Engineering and Physical Sciences Research Council (EP/X041921/1), https://www.ukri.org/councils/epsrc/, and Natural Environment Research Council (NE/X009734/1), https://www.ukri.org/councils/nerc/. NL is additionally supported by funding from the Bill & Melinda Gates Foundation (INV-064683), https://www.gatesfoundation.org. The funders had no role in study design, data collection and analysis, decision to publish, or preparation of the manuscript.

**Competing interests:** The authors have declared that no competing interests exist.

**Abbreviation:** LUCA, last universal common ancestor.

(LUCA) as all life shares the same genetic code and translation apparatus [1–5]. But the selective forces that enabled the evolution of such an intricate set of rules and machinery remain obscure.

A key theoretical challenge in understanding this transition has been how information could be maintained before the evolution of the genetic machinery. Maintaining information in a prebiotic scenario is particularly difficult because the copying-error rate had to be high in the absence of error-proofing enzymes [6]. With large numbers of copying errors, maintaining a long genetic sequence is practically impossible, but error-proofing enzymes are today invariably encoded by long genetic sequences. This is Eigen's paradox: no error-proofing enzymes without long sequences, no long sequences without error-proofing enzymes [7,8]. While Eigen's paradox could be escaped if information is contained in groups of short sequences rather than one long one, the persistence of such groups is restricted by competition for resources (such as monomers or catalysts for copying) leading to competitive exclusion. The question of how such sequences arose in the first place is also unresolved. Selection acting on RNA strands in solution favors those that are small and fast-replicating, "Spiegelman's little monsters", rather than sequences encoding other functions such as metabolism [9].

Various models of the origin of heredity have addressed the problem of information maintenance [10–13]. Among these, the simplest and most parsimonious proposal is for early sequences to evolve within dividing protocells, a solution advanced by the stochastic corrector model [14]. This model solves the problem of information maintenance through selection at the level of the protocell. Fitness is highest if the protocell maintains a specified ratio of replicator types, but declines to zero if one of the essential replicator populations falls extinct. The proliferation of cells with the optimal balance of replicator sequences allows multiple sequences to co-exist, increasing the information content that can be maintained in a protocell. In contrast, selfish fast-replicating sequences are restricted by selection at the level of the protocell, as selfish replicators copy themselves at the expense of other sequences needed for protocell division. Selection at higher levels, such as the whole cell, to overcome selfish replicators is seen as a fundamental underpinning of Major Transitions in Evolution [8].

But some questions were not addressed by the stochastic corrector and related models of early heredity [11–13,15]. These models provide a basis for the maintenance of information in protocells, but leave open the question of how protocells acquired functional sequences in the first place, or what functions these first sequences encoded. The model we present in this paper seeks to span this gap by considering how protocells first acquired sequences that enhanced the two fundamental functions enabling evolution: differential fitness and heredity [16]. In protocells, differential fitness could plausibly emerge from sequences that enhance protocell growth, for example, by encoding peptides that catalyze $CO_2$ fixation. Heredity, in turn, relies on copying linked to translation, which allows catalytic functions to be transmitted over generations. These traits inform the longstanding debate over whether metabolism or information arose first [17–21]. While our model focuses

on protocells that already have a rudimentary metabolism capable of producing amino acids and nucleotides, and which also contain heritable RNA sequences generated by random polymerization, we specifically address whether the first RNA sequences promoted metabolism or information.

We assume that protometabolic flux is catalyzed by metal ions and 'naked' cofactors (see Discussion and conclusions), which means that better $CO_2$ fixation would drive growth by generating more monomers, including fatty acids (for protocell membranes), amino acids (for peptides), and nucleotides (for RNA). The assumption of functional peptides requires the inheritance of RNA sequences that are both copied and 'translated' into catalytic peptides, so the fastest growing protocells propagate their more functional RNA. At the origin of heredity, RNA sequences must have arisen by chance and given rise to a diverse pool of translational products. We assume the most rudimentary form of translation followed simple rules based on hydrophobicity, grounded in patterns in the genetic code which suggest that interactions between amino acids and cognate nucleobases were key in early evolution [22,23]. These behaviors are determined by simple physical chemistry—folding, partitioning, and binding to catalytic cofactors emerge spontaneously from polymer composition [24–27], which is inherited from RNA templates.

The likelihood that protocells evolved peptides conferring fitness would have depended on whether they were encoded by short or long sequences and by simple or more specific motifs. This raises the question of which of the two critical functions—growth or information—was more likely to arise first and permit the evolutionary expansion of protocells. To tackle this question, we develop a model of protocells with rudimentary heredity. We assume that RNA sequences can be copied and loosely translated into peptide sequences with enzyme functions that enhance either growth or information propagation. Specifically, the model tracks protocells containing random sequences that can evolve towards two explicit functions: catalysis of $CO_2$ fixation, which increases monomer production and thus cell growth, or catalysis of templated polymerization of RNA and peptides (copying and translation). The ability of a peptide to perform one of these functions depends on its relative hydrophobicity, which determines its likelihood of folding and partitioning to either the membrane or cytosol [24–26]. We assume that hydrophobic peptides tend to associate with the membrane, where they can catalyze $CO_2$ fixation [27,28], while hydrophilic peptides remain in the cytosol, where they can facilitate templated polymerization through binding to metal ions such as $Mg^{2+}$, as in modern enzymes [29–31]. We then vary the optimal peptide compositions required for each function to ask if the RNA encoding them can evolve towards optimal sequences. We emphasize this means evolution from random RNA sequences, with zero intrinsic information, to biologically meaningful sequences encoding growth and copying peptides—the emergence of genetic information.

The model shows that these simple rules enable the evolution of functional sequences from random distributions of nucleotides and peptides, and elucidates the parameter ranges that permit protocell growth. We establish a necessary condition for this early sequence evolution to occur in autotrophic protocells: sequences encoding catalysts of $CO_2$ fixation must evolve first, as they drive the addition of monomers and, in turn, increase the rate of random polymerization and the ability of protocells to explore the sequence space, allowing the later evolution of sequences that encode information supporting functions. We suggest this observation is generalizable: as a rule, growth must come first.

## Model overview

We use an individual-based model in discrete time. At each time step a fixed sequence of probabilistic events acts on the molecular contents of each protocell. Most model parameters represent probabilities used to sample the occurrence of specific events, notably monomer addition, random polymerization, copying, translation, or decay. A population of protocells is simulated, with selection acting at the protocell level through differences in growth rate and protocell division.

## Protocell contents

The model follows in discrete time the evolution of a population of protocells containing monomers and polymers of nucleotides and amino acids. A number of simplifying assumptions are made to allow general conclusions to be drawn.

Nucleotide monomers are split into two classes, purines or pyrimidines. Purines, with their double ring structure, are somewhat more hydrophobic than pyrimidines, which have a single ring [32]. This difference is captured by considering purines to be hydrophobic and pyrimidines to be hydrophilic. Likewise, amino acids are classed as hydrophobic (such as valine or alanine) or hydrophilic (such as aspartate and glutamate).

The linear sequence of polymers made of nucleotides (RNA) and amino acids (peptides) is not explicitly considered. Instead, we keep track of the length ($l$) and relative hydrophobicity ($h$) of each polymer. The relative hydrophobicity of an RNA polymer is the difference between the number of hydrophobic purine monomers ($n_R$) and hydrophilic pyrimidine monomers ($n_Y$) divided by the length of the polymer,

$$h_n = \frac{n_R - n_Y}{l},$$

(1a)

and for peptides as the difference between the number of hydrophobic amino acid monomers ($m_\varphi$) and hydrophilic amino acid monomers ($m_\zeta$) divided by the length of the polymer,

$$h_p = \frac{m_\varphi - m_\zeta}{l}.$$

(1b)

Hence, an RNA polymer composed entirely of purines has $h_n = +1$ (conversely pyrimidines $h_n = -1$), and a peptide composed entirely of hydrophobic amino acids has $h_p = +1$ (conversely $h_p = -1$).

## Protocell dynamics

At each time step, the populations of monomers and polymers within a protocell are modified by five processes, in the following order (Fig 1). (1) Nucleotide and amino acid monomers are added through $CO_2$ fixation. Flux from the first products of fixed $CO_2$ (carboxylic acids) through a nongenetically encoded protometabolism is assumed to form nucleotides and amino acids. In the model, the numbers of nucleotides ($n$) and amino acids ($m$) added are sampled from a binomial distribution with probabilities $p_n$ and $p_m$ with $n_{max}$ and $m_{max}$ trials. It is assumed that hydrophilic and hydrophobic nucleotides and amino acids are made in equal numbers. (2) Nucleotide monomers undergo polymerization with probability $p_p$, to form either random RNA dimers or to increase the length of existing RNA polymers (this step does not apply to amino acid monomers). (3) Copying and (4) translation of RNA polymers (≥2 monomers) are independent processes that happen separately. Copying occurs via base-pairing, with the resulting RNA copy having the same length ($l$) but opposite relative hydrophobicity ($-h$) to the template. This reflects the reciprocal strand having the opposite number of purines and pyrimidines to the template. Translation is assumed to be based on hydrophobicity, with the resulting peptide having the same length ($l$) and relative hydrophobicity as the RNA template (i.e., $h_p = h_n$). For both processes, all RNAs are sampled as potential templates with a probability $p_c$ for copying and $p_t$ for translation. The sampled templates are listed randomly and copied (or translated) in order until all nucleotide and amino acid monomers, respectively, are exhausted. This assumption means the rates of copying and translation are limited by monomer supply. Incomplete RNA copies and peptide transcripts are disregarded. (5) Finally, decay occurs in which all polymers (both RNA and peptides) have a probability $p_d$ of losing a monomer. This probability is weighted by the polymer length, as longer polymers have more bonds that could be broken. This assumption means that the monomer pool generated by $CO_2$ fixation is supplemented by monomers derived from polymer decay. We appreciate that this form of polymer decay is unrealistic, but from a modeling point of view it captures two important features quite simply: it limits the length of polymers and therefore the difficulty of replicating them, and it provides an additional supply of monomers for polymerization that does not depend directly on $CO_2$ fixation.

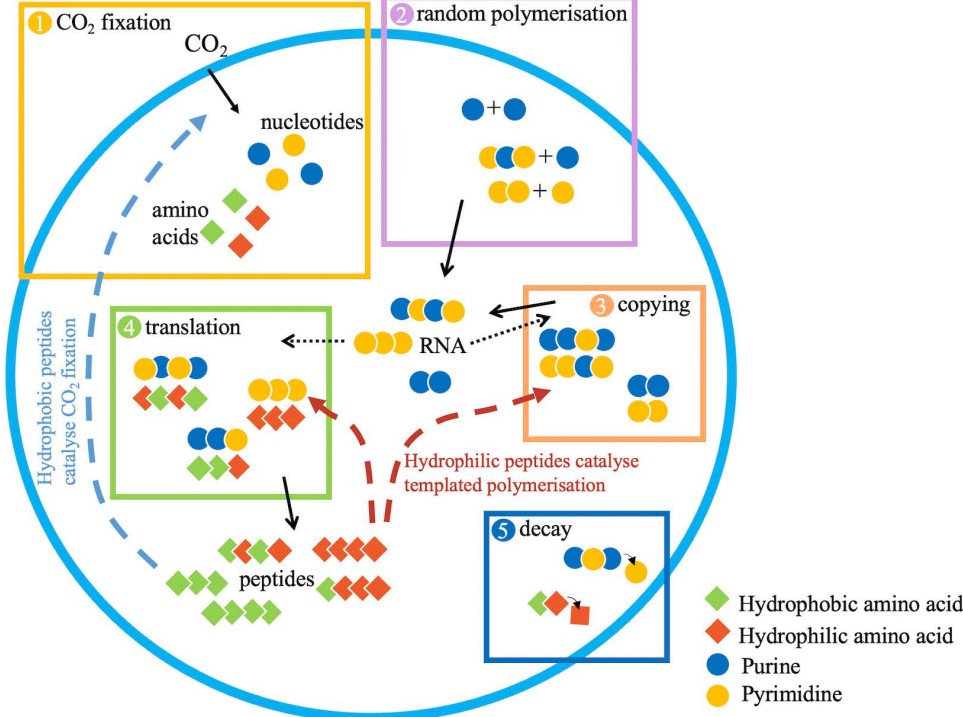

**Fig 1. Processes occurring within each protocell in the population at a single discrete time step.** Continuous arrows show polymers being produced, segmented arrows show peptide catalysis of $CO_2$ fixation and templated polymerization, and dotted arrows indicate RNA acting as templates for copying and translation. $CO_2$ fixation (or monomer addition in the heterotrophic model) and polymer decay are sources of nucleotide and amino acid monomers, while monomers are consumed by polymerization, copying, and translation. Protocells grow as a result of $CO_2$ fixation (or monomer addition) producing membrane fatty acids and divide when they reach a size threshold. Protocell division is followed by random loss of a protocell from the population (similar to a Moran process). The model tracks a population of protocells evolving under these dynamics. Sequences shown in the diagram are illustrative, and the model only keeps track of hydrophobicity and length of sequences, and not sequence order.

## Catalysis

Peptides are assumed to perform two functions. They catalyze templated polymerization, enhancing copying of RNA and translation of RNA into peptides (red dashed arrows, Fig 1). They also catalyze $CO_2$ fixation, resulting in the addition of more nucleotide, amino acids and fatty acids used for protocell growth (blue dashed arrow, Fig 1). An alternative model where monomer production is largely unlinked to protocell growth is also considered (see section on Heterotrophic growth below). Longer polymers are assumed to have greater catalytic power, which plateaus at longer lengths (Fig C in S1 Text). The parameter $k^l_{max}$ represents the maximum catalytic power and $\gamma$ represents the length at which catalytic power is at half maximum. Higher values of $\gamma$ indicate that the optimal catalytic power is achieved by longer peptides, so we use this parameter as a proxy for the optimal length of a sequence.

Hydrophobic peptides are assumed to preferentially partition to the membrane [26], where they catalyze $CO_2$ fixation, similar to the Energy-converting hydrogenase in methanogenic Archaea [33,34]. In contrast, hydrophilic peptides tend to remain in the cytosol, where they bind metal ions and catalyze the templated polymerization of RNA and peptides. Given the deep differences between RNA and peptide synthesis, it might seem unjustified to group them as a single function performed by the same class of catalysts, but the RNA polymerase and ribosome both require $Mg^{2+}$ for their catalytic function [35–37], and both processes condense activated monomers by eliminating pyrophosphate, so the physical chemistry is in part equivalent.

The relationship between peptide relative hydrophobicity and catalysis is modeled as a Gaussian curve with an optimum catalytic power at a specific relative hydrophobicity (Fig D in S1 Text). The relative hydrophobicity that result in optimal catalysis of $CO_2$ fixation and templated polymerization is given by the constants $\beta_{fix}$ and $\beta_{pol}$ respectively (Fig D in S1 Text). The $\beta$ parameters were chosen so that the peaks of these curves do not overlap—so, some peptides might have low catalytic activity for either function, but no peptides have high catalytic activity for both functions.

### Growth, division, and selection

Protocells are assumed to grow at a rate proportional to that of $CO_2$ fixation. This follows from the assumption that the rate of fatty-acid synthesis (constituting new protocell membrane) is necessarily proportional to that of $CO_2$ fixation in autotrophic protocells. The same logic applies to nucleotides and amino acids, which are produced at a rate proportional to $CO_2$ fixation. When a protocell reaches an arbitrary size threshold, set by the number of fatty acids in the protocell, we assume it divides, and one other protocell is deleted at random from the population, in a similar process to a Moran model [38]. The RNA and peptide polymers and monomers are divided between the two daughter cells with proportions $x$ and $1 - x$, where $x$ is sampled from a normal distribution truncated between 0 and 1, with mean 0.5 and standard deviation 0.2.

In each simulation, 100 protocells are allowed to evolve for 10,000 time steps, where a time step is one 'turn' of the model (i.e., the 5 sequential processes in Fig 1). At the end of each time step, a protocell can divide if it has reached the threshold number of fatty acids for replication ($S$). We investigate how the rate of protocell division changes for different parameters and features of the model. In the model, selection arises simply from the growth rate, favoring faster rates of protocell growth. A full description of the model is given in the Supporting information.

### Parameter values

A simplified model where optimal catalytic sequences are symmetrical was used to investigate the effect of each parameter on the dynamics. In the simplified model, the relationship between the length and the relative hydrophobicity of amino acid polymers on catalytic power is the same for $CO_2$ fixation and templated polymerization. This leads to the unrealistic feature that a "Watson" nucleotide polymer that favors $CO_2$ fixation when copied automatically generates a "Crick" nucleotide polymer that favors templated polymerization to the same extent. This model is only used to set parameter values that lead to protocell growth, used as the base values in Table 1. We report variation in these parameters in the Supporting information. For instance, increasing the baseline probabilities of copying and translation ($p_c$, $p_t$) and decreasing the probability of decay ($p_d$) leads to higher catalytic rates and protocell growth (S4–S6 Figs).

### Heterotrophic growth

An alternative model where protocells are heterotrophic is also considered. In this model, protocell growth and monomer addition are unlinked. Here, hydrophobic catalysts increase the number of fatty acids used for protocell growth but do not affect the intake of nucleotides or amino acids. Both nucleotides and amino acids are taken up from the environment at constant rates $p_n$ and $p_{aa}$, respectively. Note that the $CO_2$ fixation catalyst was replaced with a growth catalyst rather than a catalyst enhancing intake of monomers because a function that directly links to protocell division is still necessary. Otherwise, the lack of a growth catalyst would render selection at the level of the protocell impossible.

### Code and simulations

We implemented the model using Matlab (vR2021a, The Mathworks, MA, USA). The Matlab scripts are available on Github: https://github.com/raquelnpalmeira/first_growth_then_information and archived on Zenodo: https://doi.org/10.5281/zenodo.18940155.

**Table 1. Parameters used in simulations (unless otherwise stated).**

| Parameter | Symbol | Value |
|---|---|---|
| Maximum number of nucleotide monomers added per time step | $n_{max}$ | $10^4$ |
| Maximum number of amino acid monomers added per time step | $m_{max}$ | $10^4$ |
| Probability of nucleotide monomer addition | $p_n$ | 0.001 |
| Probability of amino acids monomer addition | $p_m$ | 0.001 |
| Probability of random polymerization of nucleotides | $p_p$ | 0.01 |
| Probability of polymer decay | $p_d$ | 0.001 |
| Probability of copying of nucleotide polymers | $p_c$ | 0.1 |
| Probability of translation of nucleotide polymers into peptides | $p_t$ | 0.01 |
| Catalytic constant for catalysis of $CO_2$ fixation based on hydrophobicity (for symmetrical model) | $\alpha_{fix}$ | 4 |
| Catalytic constant for catalysis of templating based on hydrophobicity (for symmetrical model) | $\alpha_{pol}$ | 4 |
| Catalytic constant for catalysis of $CO_2$ fixation based on hydrophobicity (for nonlinear model) | $\beta_{fix}$ | 4 |
| Catalytic constant for catalysis of templated polymerization based on hydrophobicity (for nonlinear model) | $\beta_{pol}.$ | 4 |
| Catalytic constant for catalysis of $CO_2$ fixation based on length | $\gamma_{fix}$ | 6 |
| Catalytic constant for catalysis of templated polymerization based on length | $\gamma_{pol}$ | 6 |
| Maximum catalytic value for catalysis based on length | $k^l_{max}$ | 3 |
| Number of protocells in the population | $c$ | 100 |
| Threshold size for division | $S$ | $10^4$ |

## Results

### Evolution requires $CO_2$ fixation to be easier than copying

Reliable evolution of protocells with high rates of division is possible when the optimal sequences for $CO_2$ fixation are easier to generate at random than those for templated polymerization. Specifically, when $CO_2$ fixation catalysts are shorter ($\gamma_{fix} < \gamma_{pol}$) and weakly hydrophobic ($|\beta_{fix}| < |\beta_{pol}|$) they arise more easily by chance and provide an initial selective advantage. In this case, the rate of protocell division quickly and consistently evolves to a high level (blue line, Fig 2A). The increased rate of $CO_2$ fixation results in more monomers being generated per time step, increasing the probability of random polymerization. This allows protocells to explore sequence space and generate the less probable RNA sequences that encode catalysts for templated polymerization. Selection then gives rise to protocells with RNA polymers that encode both catalytic optima, one for $CO_2$ fixation and the other for templated polymerization (Fig 2B). This leads to a positive feedback loop, with selection favoring high rates of protocell growth (Fig 2A). This result exposes a previously overlooked feedback between autotrophic metabolism and evolvability. Metabolic flux leads to more monomers which allow for greater exploration of the RNA sequence space.

In contrast, when RNA strands coding for catalysis of templated polymerization are shorter ($\gamma_{fix} > \gamma_{pol}$) and weakly hydrophilic ($|\beta_{fix}| > |\beta_{pol}|$), protocells with high rates of division never evolve (orange line, Fig 2A). In this case, catalysts for templated polymerization arise at random more easily and increase the rate of copying and translation. But that makes it less likely that longer sequences with more extreme hydrophobicity needed to catalyze $CO_2$ fixation arise by chance. The distribution of RNA strands under this condition is little better than random (Fig 2C). Without the production of RNA strands encoding catalysts of $CO_2$ fixation the monomer supply remains limited, restricting protocell growth.

We also explored simulations in which one of the parameters (RNA length or relative hydrophobicity) was more extreme for $CO_2$ fixation while the other was more extreme for templated polymerization. Under these intermediate conditions, neither one sequence nor the other is favored to arise at random more strongly than the other. Evolution towards higher

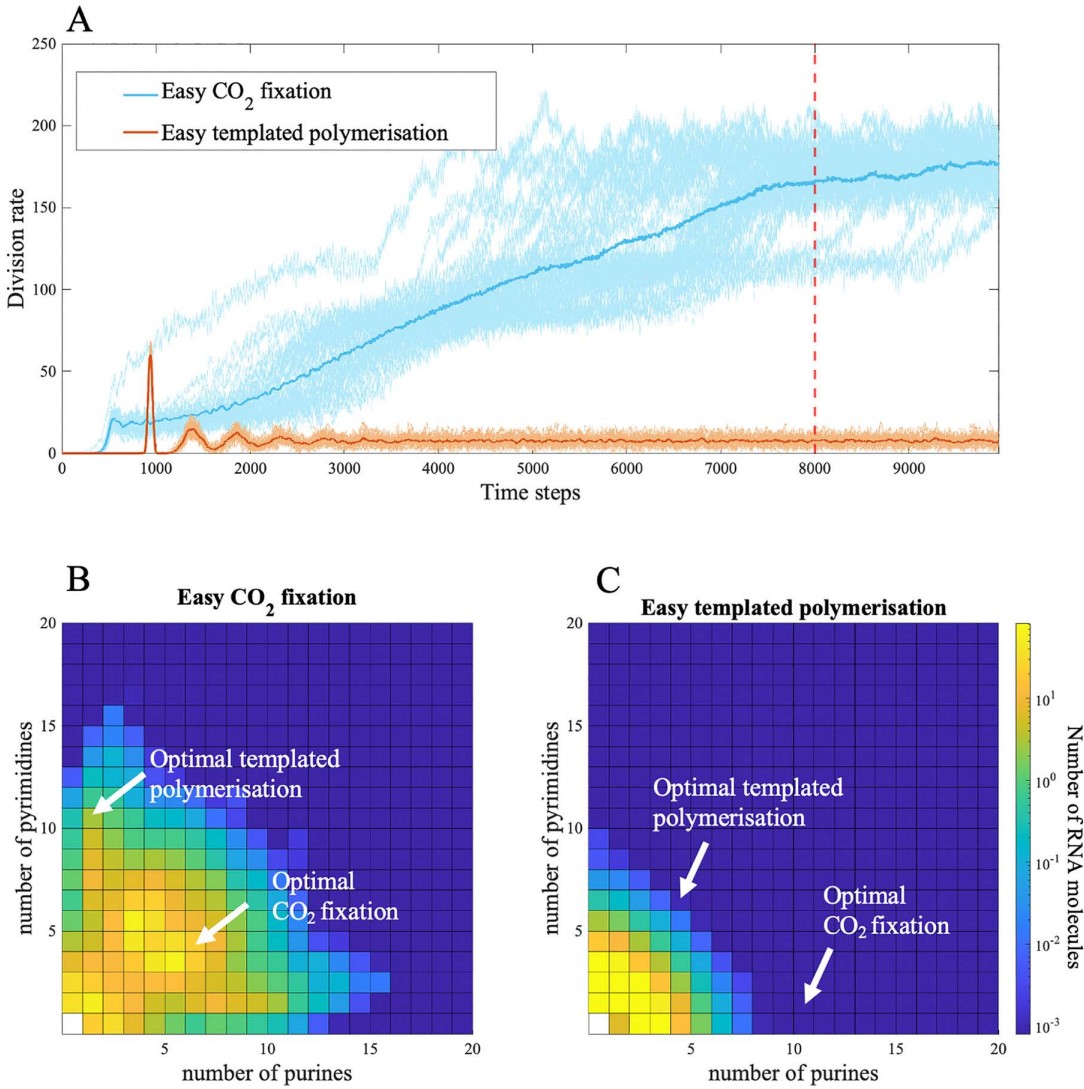

**Fig 2. Evolution of protocells with different catalytic optima for $CO_2$ fixation and templated polymerization.** Panel (A) shows the moving sum of protocell divisions over 50 time steps, and panels (B–C) show the average RNA compositions of protocells at time step 8,000. Catalytic rates ($k_l$ and $k_h$) depend on the optimal catalytic length ($\gamma$) and hydrophobicity ($\beta$) for $CO_2$ fixation ($\gamma_{fix}$, $\beta_{fix}$) and for templated polymerization ($\gamma_{pol}$, $\beta_{pol}$). We compare two contrasting scenarios in which either $CO_2$ fixation or templated polymerization catalysts are easier to achieve at random, due to differences in their optimal $\gamma$ and $\beta$. values. In panel (B) (blue line in panel a), $CO_2$ fixation catalysts are easier to evolve, with $\gamma_{fix} = 6$, $\gamma_{pol} = 8$, $\beta_{fix} = 0.2$, and $\beta_{pol} = -0.8$. In panel (C) (orange line in panel a), templated polymerization catalysts are easi to evolve, with $\gamma_{fix} = 8$, $\gamma_{pol} = 6$, $\beta_{fix} = 0.8$, and $\beta_{pol} = -0.2$. Each square in the heatmaps shows the log count of RNA molecules with a given number of purines and pyrimidines. When $CO_2$ fixation catalysts are easier to form at random, protocell division rates increase (blue line in panel A) and RNA distributions become centered on both catalytic optima (B); when templated polymerization is favored, division rates remain low (orange line in panel A) and RNA distributions stay close to random (C). All other parameters are listed in Table 1. The data and scripts used to generate this figure are available in the GitHub repository archived on Zenodo (https://doi.org/10.5281/zenodo.18940155, folder Fig 2).

protocell division rates can occur but at a low frequency (1%–3% of simulations, S5 Fig). This occurs in the unlikely case that RNA strands coding for $CO_2$ fixation do form and initiate a selective advantage through greater input of monomers. Only then is it beneficial for the protocell lineage to evolve better templating (i.e., copying and translation) as well.

## Decoupling growth and monomer supply relaxes constraints on growth

The results above depend on the assumption that protocells are autotrophic, making organics by reducing $CO_2$. In the model, the rate of protocell growth is proportional to the rate of monomer addition. This favors sequences that raise protocell fitness by increasing monomer input, with the downstream effect that sequence space is better explored, enhancing catalytic functions.

If the link between protocell growth and monomer availability is broken, these feedbacks no longer apply as strongly. If monomers are delivered at high rates from an exogenous 'soup' (implying a heterotrophic origin), then sequence space will inevitably be explored, albeit 'easy growth' is still slightly favored over 'easy templated polymerization' (Fig 3A). Protocells evolve towards higher division rates irrespective of whether optimal catalytic sequences are easier to generate at random for $CO_2$ fixation ($\gamma_{fix} < \gamma_{pol}$ and $|\beta_{fix}| < |\beta_{pol}|$; Fig 3B) or templated polymerization ($\gamma_{fix} > \gamma_{pol}$ and $|\beta_{fix}| > |\beta_{pol}|$; Fig 3C). The high rate of monomer input (specifically amino acids and nucleotides) means that even if catalysis of templated polymerization evolves first, this does not deplete monomers sufficiently to hinder the production by random nucleotide polymerization of RNA molecules that code for peptides catalyzing growth. In other words, growth is still required for the evolution of functions beyond RNA replication, but is not needed for the cell to explore sequence space through random polymerization.

Critically, this latter observation only holds true if monomer supply is high enough for protocells to freely explore sequence space, such that RNA copying and random polymerization do not compete for monomers (Fig 4, $p_n = p_{aa} = 0.01$). If monomer supply is limiting, then protocells never sharply increase their division rates (Fig 4, $p_n = p_{aa} = 0.001, 0.0001,$ and $0.00001$). Specifically, at intermediate rates of monomer supply, roughly equal to the baseline monomer supply for the autotrophic model growth remains low in all simulations (Fig 4, $p_n = p_{aa} = 0.001$). While the model was not intended to directly compare autotrophic versus heterotrophic origins, it is worth noting that heterotrophic growth rates are substantially lower than autotrophic growth rates. That happens because selection for protocell growth in the autotrophic model also leads to increased monomer synthesis, and more monomers generate more catalysts, forming a positive feedback loop. This loop is not possible under the heterotrophic model.

## Random polymerization does not rescue growth when $CO_2$ fixation is disfavored

Considering again an autotrophic model (where growth and monomer supply are fully linked), a higher probability of random polymerization potentially could alleviate the requirement for $CO_2$ fixation to evolve more easily than templated polymerization. To test this idea, the random polymerization of nucleotides ($p_p$) was raised in simulations where catalysis of $CO_2$ fixation was less likely to arise by chance than templated polymerization ($\gamma_{fix} > \gamma_{pol}$ and $|\beta_{fix}| > |\beta_{pol}|$). The rate of protocell division does increase slightly at higher probability of random polymerization ($p_p = 0.01$–$0.08$; Fig 5A), as catalysis of $CO_2$ fixation arises more often by chance. However, further rises in the probability of random polymerization do not increase the rate of protocell division, and overall there is little fitness advantage ($p_p = 0.12$; Fig 5A). This is because higher values of $p_p$ also make random polymerization more dominant, which dilutes any high-fitness RNA sequences with random noise (Fig 5B–5E). As a result, the system fails to generate the positive feedback loop described earlier, in which catalysis of $CO_2$ fixation enhances monomer supply, which in turn improves copying, again amplifying $CO_2$ fixation. Without this feedback loop, RNA distributions never center around the catalytic optima (Fig 5B–5E) and protocell divisions always remain low (Fig 5A)—unlike cases where $CO_2$ fixation is favored and distributions become centered on functional sequences (Fig 2B).

In the complementary case, where catalysis of $CO_2$ fixation is favored ($\gamma_{fix} < \gamma_{pol}$ and $|\beta_{fix}| < |\beta_{pol}|$), protocell division rates fall modestly but consistently with increasing probabilities of random polymerization (S6 Fig). The same dynamics apply here: random polymerization reduces the relative advantage of templated copying and disrupts the maintenance of catalytic RNAs. Although protocell growth is still possible, the efficiency of selection is reduced by noise.

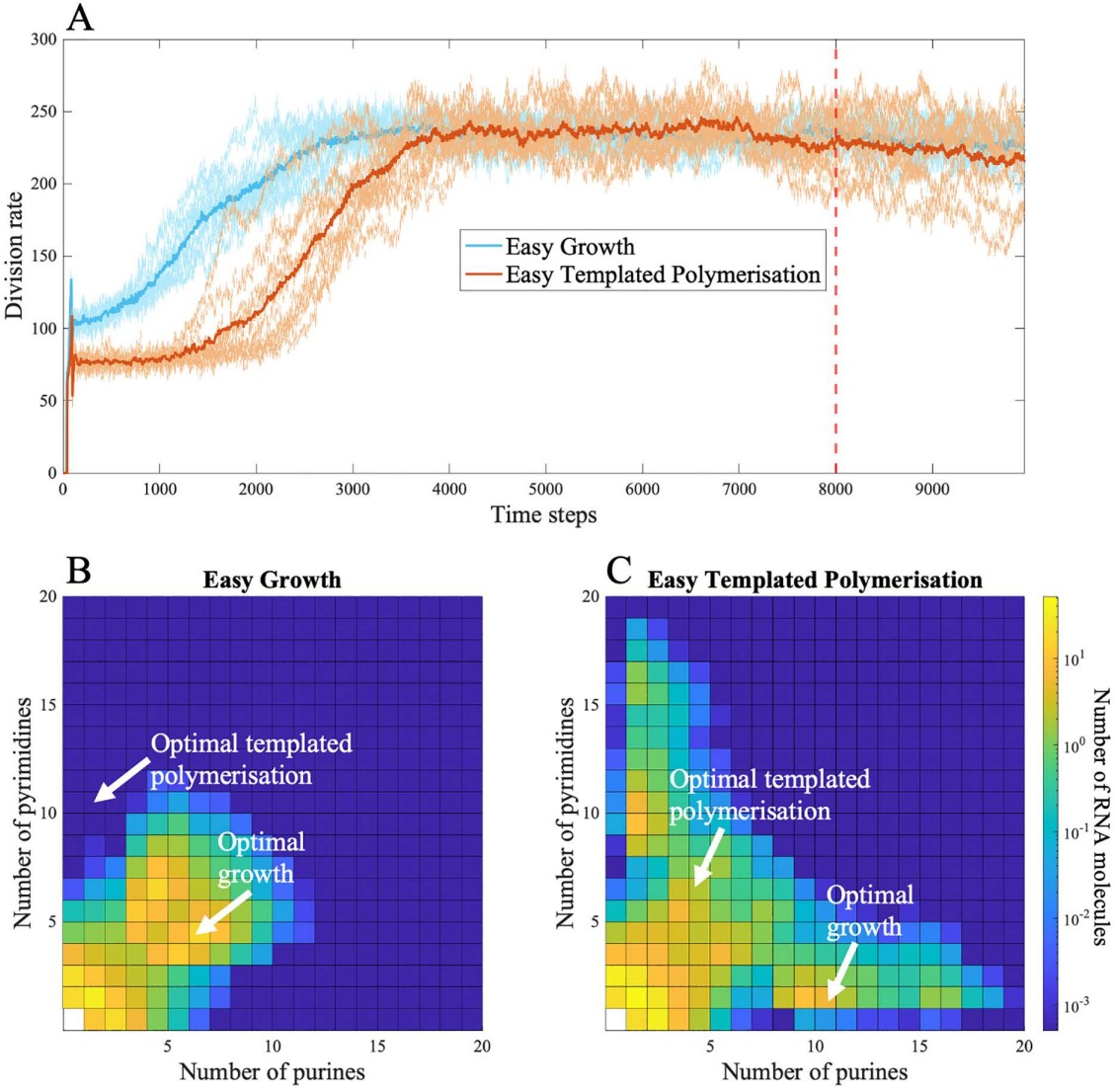

**Fig 3. Evolution of protocells with different catalytic optima for growth and templated polymerization, when growth catalysis is unlinked to monomer supply and $p_n = p_{aa} = 0.1$. Panel (A) shows the moving sum of protocell divisions over 50 time steps, and panels (B–C) show the average RNA compositions of protocells at time step 8,000.** Catalytic rates ($k_l$ and $k_h$) depend on the optimal catalytic length ($\gamma$) and hydrophobicity ($\beta$) for growth ($\gamma_g$, $\beta_g$) and for templated polymerization ($\gamma_{pol}$, $\beta_{pol}$). We compare two contrasting scenarios in which either growth or templated polymerization catalysts are easier to achieve at random, due to differences in their optimal $\gamma$ and $\beta$ values. When growth catalysis is decoupled from monomer supply and monomer availability is high, protocells evolve high division rates irrespective of whether catalysts for growth (blue line) or templated polymerization (orange line) are easier to generate at random. In panel (B) (blue line in panel A), growth catalysts are easier to evolve, with $\gamma_g = 6$, $\gamma_{pol} = 8$, $\beta_g = 0.2$, and $\beta_{pol} = -0.8$. In panel (C) (orange line in panel a), templated polymerization catalysts are easier to evolve, with $\gamma_g = 8$, $\gamma_{pol} = 6$, $\beta_g = 0.8$, and $\beta_{pol} = -0.2$. Each square in the heatmaps shows the log count of RNA molecules with a given number of purines and pyrimidines. All other parameters are listed in Table 1. The data and scripts used to generate this figure are available in the GitHub repository archived on Zenodo (https://doi.org/10.5281/zenodo.18940155, folder Fig 3).

## Discussion and conclusions

The backdrop to the question of the origin of genetic heredity and evolution can be framed in terms of the basic requirements for evolution by natural selection [16]. Given rudimentary heredity and natural variation produced by stochasticity,

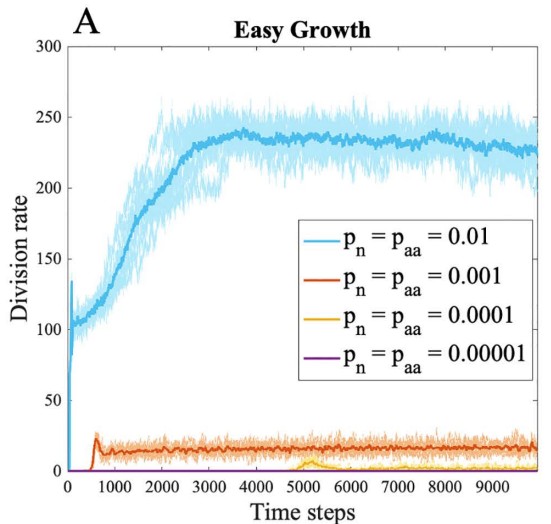
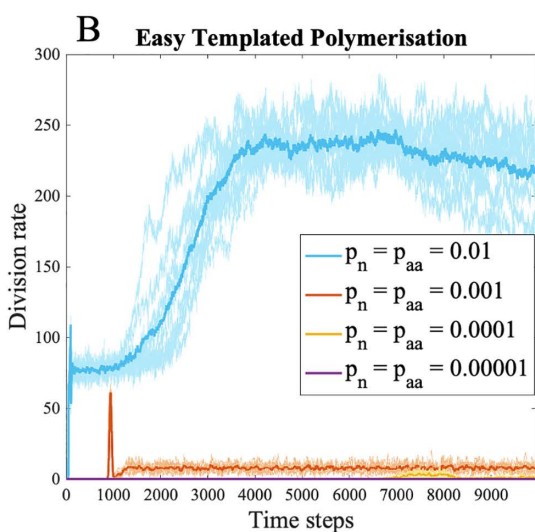

**Fig 4. Decoupling growth and monomer supply.** Evolution of protocells when growth catalysis is unlinked to monomer supply with variable rates of nucleotide and amino acid monomer addition ($p_n$ and $p_{aa}$). Panels (A) and (B) show the moving sum of protocell divisions over 50 time steps for decreasing rates of monomer addition ($p_n = p_{aa} = 0.01$, 0.001, 0.0001 and 0.00001), shown in blue, orange, yellow, and purple, respectively. Panel (A) shows the scenario when growth catalysts are easier to evolve ($\gamma_g = 6$, $\gamma_{pol} = 8$, $\beta_g = 0.2$, and $\beta_{pol} = -0.8$). Panel (B) shows the scenario when growth catalysts are easier to evolve ($\gamma_g = 8$, $\gamma_{pol} = 6$, $\beta_g = 0.8$, and $\beta_{pol} = -0.2$). At monomer supply rates comparable to those driving robust growth in the autotrophic model ($p_n = p_{aa} = 0.001$), protocell division remains low, and further reductions in monomer availability halt growth altogether. All other parameters are listed in Table 1. The data and scripts used to generate this figure are available in the GitHub repository archived on Zenodo (https://doi.org/10.5281/zenodo.18940155, folder Fig 4).

what should be favored first: differential fitness (protocell growth) or reliable heritability (templated polymerization)? This might appear to be a classic 'chicken-and-egg' dilemma, but the mechanistic details of protocell function provide insight. In the model, the two functions are encoded by distinct sequences. An important outcome is that sequences encoding $CO_2$ fixation for growth must be synthesized more readily, and evolve before those encoding templated polymerization. In other words, differential fitness must arise first, and reliable heritability second. The reason lies in the fact that, in autotrophic protocells, 'protocell fitness' is intrinsically linked to the production of new sequences—$CO_2$ fixation produces membrane fatty acids but also nucleotides and amino acids. If the best catalysts for $CO_2$ fixation are short and have modest specificity of hydrophobicity, they will be more easily generated at random and promote higher rates of protocell division (Fig 2A). The resulting increase in $CO_2$ fixation promotes monomer production, which in turn drives higher rates of polymerization, enabling protocells to explore RNA sequence space, and eventually producing longer, more specific hydrophilic catalysts that support templated polymerization (Fig 2B). The resulting positive feedback loop leads to improved copying and translation alongside $CO_2$ fixation. By contrast, if templated polymerization catalysts arise first—because they are shorter or more weakly hydrophobic—then they consume nucleotide monomers, making it harder for protocells to synthesize the longer, more hydrophobic catalysts needed for $CO_2$ fixation (Fig 2C). As a result, protocell growth never takes off. We can draw a general conclusion from this finding: differential fitness must precede reliable heritability, and growth precede copying.

This conclusion rests on the link between monomer supply and protocell growth. If the $CO_2$ fixation catalyst is replaced by a 'growth catalyst'—one that only promotes fatty-acid synthesis or scavenging—and the nucleotide and amino acid monomers are added at a steady, constant rate, the results change. This version of the model breaks the assumption of autotrophy and mimics a heterotrophic protocell in which monomer supply is externally fixed and uncoupled from RNA encoded catalysis. Under these conditions, and with a high rate of external monomer supply, either class of

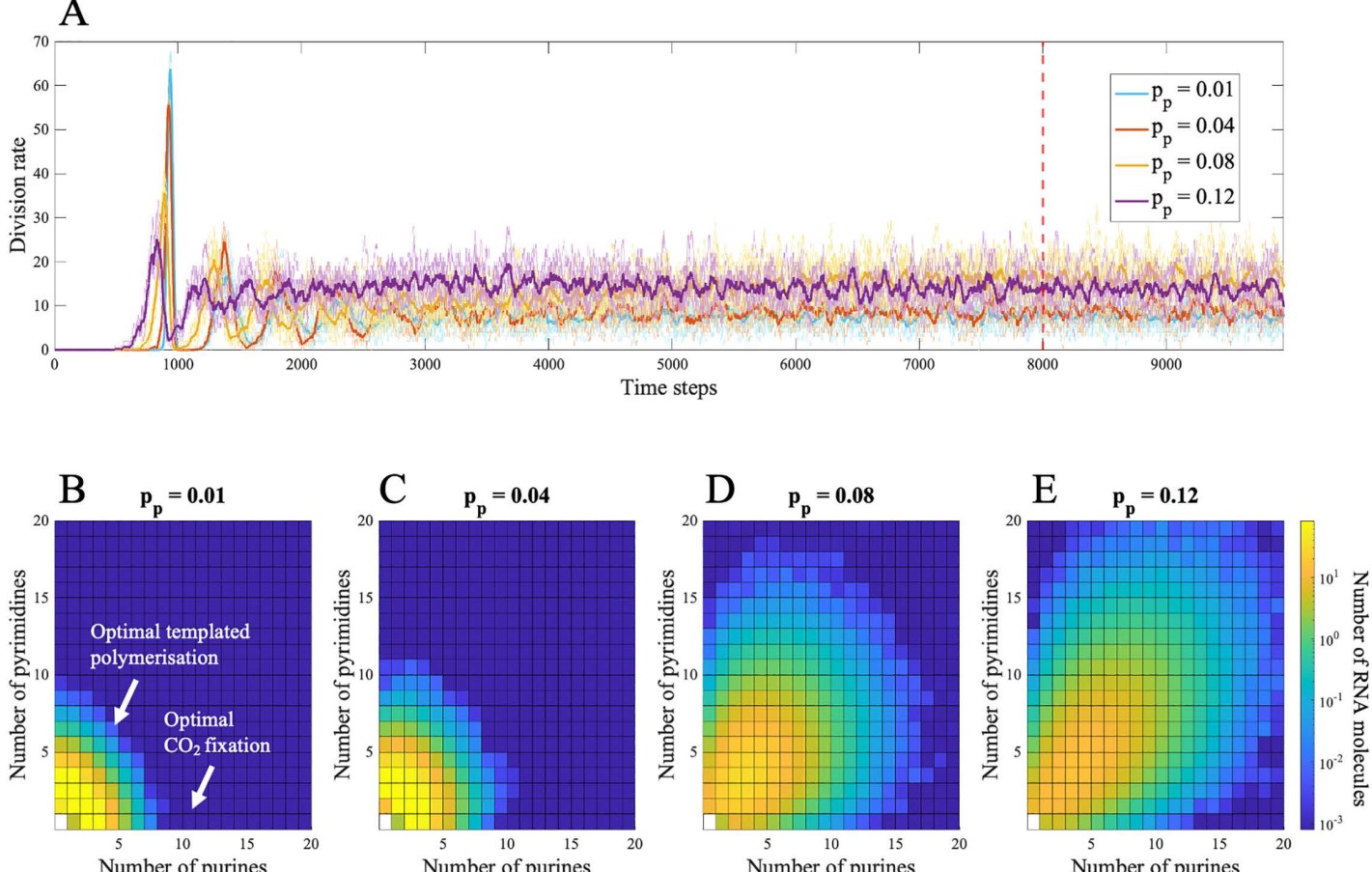

**Fig 5. The effect of varying the probability of random polymerization ($p_p$).** This is investigated when templated polymerization catalysts are easier to achieve at random ($\gamma_{fix}$ = 8, $\gamma_{pol}$ = 6, $\beta_{fix}$ = 0.8, and $\beta_{pol}$ = –0.2.). (A) Evolutionary change in the rate of protocell divisions (per 50 time steps) for $p_p$= 0.01 (blue), $p_p$= 0.04 (orange), $p_p$=0.08 (yellow), and $p_p$ = 0.12 (purple). The mean distribution of nucleotides across protocells at **t = 8,000** (red dotted line in panel (A)) is shown for (B) $p_p$ = 0.01, (C) $p_p$ = 0.04, (D) $p_p$ = 0.08 and (E) $p_p$ = 0.12. Each square in the heatmap shows the log count of RNA molecules composed of specific numbers of purines and pyrimidines. Overall, increasing $p_p$ produces only a modest increase in protocell division at low values, while higher rates of random polymerization disrupt the maintenance of functional RNA distributions by introducing noise. All other parameter values are in Table 1. The data and scripts used to generate this figure are available in the GitHub repository archived on Zenodo (https://doi.org/10.5281/zenodo.18940155, folder Fig 5).

catalysts—those for growth or for templated polymerization—can evolve first, and protocells still achieve high division rates (Fig 3A). Because monomers are no longer limiting, the evolutionary order of functions is less constrained. But in the more likely case that external monomer supply is steady but limiting—equivalent to baseline monomer availability in the autotrophic model—protocells evolve more reliably to slightly higher division rates when growth catalysts are easier to evolve at random (Fig 4, $p_n$ = $p_{aa}$ = 0.01). If the monomer supply from the environment is even lower, protocell growth fails to take off at all (Fig 4, $p_n$ = $p_{aa}$ = 0.001). While the availability of monomers in any 'primordial soup' is hard to predict, uniformly high concentrations seem implausible. If all types of monomers were not uniformly available (which would likely be the case for hydrophobic amino acids and nucleotides), then rates of growth would collapse to those *supported* by the availability of scarcest monomer. Worse, the heterotrophic model still requires either de novo fatty-acid synthesis from $CO_2$ fixation (no easier than the other pathways assumed in the autotrophic model) or some form of differential

scavenging of fatty acids from the environment by the growth catalyst. Given the physical tendency of fatty acids to incorporate directly into fatty-acid bilayers, it is hard to see how such a catalyst could benefit some cells over others, in which case cell-level selection would not be possible. In the modeling, the heterotrophic model serves as a valuable proof of concept. A reliable form of heredity can evolve independent of which peptide function arises first but only when monomer supply is abundant and does not limit sequence formation. This condition seems implausible.

Our choice to focus on autotrophic protocells is justified by phylogenetic and experimental data. Phylogenetic reconstructions show that LUCA was likely chemolithoautotrophic converting $CO_2$ and $H_2$ into organics [1,5,39], and experimental work has shown entire sections of autotrophic metabolism are possible under plausible prebiotic conditions [28,40–46]. Patterns in the genetic code provide further support: the first base of the codon correlates with the biosynthetic distance of the amino acid from $CO_2$ fixation, with amino acids closer to the metabolic core encoded by purines and more distal ones by pyrimidines [23]. These observations strongly suggest that the genetic code arose in the context of an expanding autotrophic metabolism rooted in $CO_2$ fixation.

A seemingly alternative way autotrophic protocells could have explored large regions of sequence space would be high rates of random polymerization. This increases the likelihood of generating functional sequences by chance, potentially bypassing the need for $CO_2$ fixation catalysts to arise first. To test this, the probability of random polymerization ($p_p$) was varied. However, the effect of increasing $p_p$ was limited or detrimental. Increase in random polymerization allows easier exploration of parameter space but it also increases competition for monomers needed for copying. This reduces the amplification of beneficial sequences thereby stalling growth (Figs 5A and S6). Protocells are only able to accumulate functional sequences when copying is capable of outcompeting noise (i.e., random polymerization). This reveals an important asymmetry. Higher values of random polymerization increase the rate at which random sequences form, but does not feed through to an increase the total monomer pool. In contrast, increasing monomer input through raised $CO_2$ fixation expands the pool of available monomers and allows copying, once established, to dominate over random processes. In this way, growth enables information not just by increasing exploration, but by increasing the supply of monomers that copying can exploit, allowing beneficial sequences to be selectively amplified.

Although increased random polymerization did not alter the overall pattern of results, it did decrease the ability of protocells to retain sequence distributions centered around catalytic optima (Fig 5C–5E). Some degree of random polymerization is necessary to explore sequence space, but excessive rates reduce the efficiency of selection and hinder the maintenance of functional sequences. This is a standard result of population genetics: noise reduces the efficacy of selection [47]. It also seems plausible that templated polymerization occurs at a higher rate than random polymerization. De novo polymerization of nucleotides in water is difficult. The only successful attempts have used wet-dry cycles [48], eutectic freezing [49], thermal gradients [50], and basalt rock glasses [51], but the relevance of these conditions to protocells is ambiguous as both RNA and peptide polymerization involves monomer activation and elimination of pyrophosphate [52,53]. There have been fewer attempts to elucidate the conditions needed for templated polymerization of nucleotides. This has also proved difficult, and the only successful attempts were achieved using 'pre-condensed' monomers (nucleoside phosphorimidazoles), which are not consonant with life [54]. While both reactions are clearly difficult, templated polymerization should enable monomers to bind to the template through hydrogen bonding and stacking interactions [55,56], which partially immobilizes them, increasing the likelihood of condensation to form a polymer. It is then credible that modest catalytic enhancement—whether by cofactors, ribozymes, or short peptides, simple precursors of the $Mg^{2+}$-dependent RNA polymerase, as modeled here—could have tipped the balance in favor of copying over random polymerization.

There are good reasons to think that $CO_2$ fixation can indeed evolve more easily than templated polymerization, as there are many ways to enhance $CO_2$ fixation. The phrase '$CO_2$ fixation' implicitly includes all intermediary metabolism between the initial fixing of $CO_2$ through to the synthesis of fatty acids, amino acids, and nucleotides. Catalysts of $CO_2$ fixation, therefore, include any catalysts that drive flux through intermediary metabolism, and consequently cell growth. A caveat to this is

that flux through the metabolic network must be balanced. Our earlier work shows that strong catalysis of a single pathway (e.g., the synthesis of sugars or amino acids) diverts flux down one pathway at the expense of others, unbalancing metabolism and slowing down protocell growth and division [57]. As such, catalysts of $CO_2$ fixation include promiscuous 'naked' cofactors (which function in the absence of an enzyme, just slower) that catalyze various metabolic pathways simultaneously, such as the transfer of hydride ions by NADH or equivalents [57–59]. $CO_2$ fixation can be facilitated by NADH in acetogenic bacteria [60], by [4Fe-4S] clusters in Ech or ferredoxin [33,61,62], by Mo or Ni in CODH [63,64], and by nucleotide cofactors such as pterins and folates in methanogens and acetogens [39,65,66]. Conversely, RNA polymerization involves only one specific repeated reaction, and all polymerases have a conserved functional core [67].

To uncover the conditions favoring this evolutionary transition to the emergence of genetic information, we drew on patterns in the genetic code, which predict that translation emerged through direct physical interactions between amino acids and the RNA bases that code for them. Associations guiding codon assignment have been documented since the 1960s [68–71], and amount to the so-called 'code within the codons' [22,72]. The clearest pattern links the hydrophobicity of the RNA anticodon middle base with that of the cognate amino acid [23]. This connection suggests that early translation was based on direct interactions between anticodon bases and amino acids with similar hydrophobicity, or more subtly, partition energy [73]. The existence of direct physical interactions is supported by miscellaneous experimental and computational work [71,74–77], albeit this has remained controversial [78]. Recently, systematic molecular dynamics simulations showed that half of all proteinogenic amino acids interact most strongly with the cognate middle base of the anticodon, and this general trend was confirmed by NMR [79]. If these interactions were at the heart of coding, then early genetic information would have been simple and coarse grained, producing what Woese termed 'statistical proteins' [69]. Coding would not have been based on the sequence of specific nucleotides, but on their general hydrophobicity (or as noted, related interactions). Because purines are more hydrophobic than pyrimidines, a succession of purines in RNA would template a relatively hydrophobic peptide through direct interactions between amino acids and cognate bases. This assumption underpins our model.

The assumption that early translation was based on hydrophobicity provides a bridge between the structure of the genetic code and the emergence of functional peptides in an autotrophic protocell context. In principle, the same logic could apply to a model without peptides, in which RNA sequences alone acted as catalysts. If RNA sequences folded to catalyze both $CO_2$ fixation and their own replication, selection would still favor the same ordering of functions. But this scenario comes with an unhelpful trade-off: ribozymes that fold into complex catalytic structures are likely to be harder to copy, as their complex secondary structure needs to be unpicked, creating a tension between their catalytic performance and replicability. The current model does not need to account for that trade-off, as peptides are responsible for all functions. In any case, even if these functions are carried out by ribozymes rather than peptides, the overall result would still hold: $CO_2$ fixation needs to arise before reliable heredity, whether catalyzed by peptides or RNA, so long as it fuels monomer production and supports sequence exploration and selective amplification. Of course, the assumption that $CO_2$ fixation (or other core metabolic functions) is catalyzed by ribozymes rather than enzymes is ungrounded in empirical observation (there are no known examples in life) and even less helpfully, defers the question of how proteins came to replace ribozymes, which is solved simply here through direct physical interactions. We have therefore focused on loosely templated translation and the evolution of Woeseian 'statistical' proteins. It is worth noting that the system outlined here, based on direct physical interactions, is relatively robust to errors, as a hydrophobic amino acid is more likely to be replaced by an equivalent hydrophobic amino acid than its polar opposite, which accounts simply for the apparent optimization of the genetic code [80].

For simplicity, neither version of the model considers loss or gain of polymers from the environment. Full polymers were lost only when a cell divides and another in the population was deleted at random. This assumption is consistent with the fact that, in modern cells, specific transporters are needed to import nucleosides and amino acids [81]. Even so, including random polymer loss or gain, for instance, during protocell division, would be an extra source of stochasticity, which

would reinforce the conclusion that copying needs to be stronger than random polymerization to decrease the noise in the system. It would therefore be even more important to reliably produce catalysts for $CO_2$ fixation, as these could be lost to the environment at any point. A further simplification is the model of copying. This does not consider incomplete copying, which would increase the abundance of short polymers and reduce the fraction of polymers that reach functional length. That in turn would make functional sequences harder to reach and increase stochasticity, but would not be expected to change the qualitative outcome. The same applies to error rates in either copying or translation, which increase the noise in the system, placing a greater premium on catalysts of copying rather than random polymerization.

The model also assumes that polymer decay occurs through loss of monomers from the ends of the polymers. This assumption is consistent with the empirical observation that bacterial and archaeal exonucleases primarily initiate RNA degradation from the 3′→5′ end [82–85]. While 3′→5′ decay is not true of prebiotic chemistry (where attack from the 2′ hydroxyl can occur anywhere along the chain) degradation may still occur from the 3′ end if it is not buried in a folded RNA [86], or if any random polymers exerted limited exonuclease activity from the 3′ end. In any case, the assumption of RNA decay through monomer loss is conservative because internal breakage would generate two shorter polymers rather than a polymer and a free monomer. That in turn would increase the abundance of shorter polymers (increasing noise) and exacerbate the limitation of monomers, placing even greater emphasis on growth first to enhance monomer availability.

If genetic heredity began with biases in peptide populations within protocells, the results presented here clarify the conditions under which nucleotide polymers could have evolved robustly. A critical requirement is that copying must outcompete random polymerization, so that useful sequences are maintained rather than lost to noise. The model also reveals the order in which key functions are likely to have emerged. Sequences enabling $CO_2$ fixation, which amplify monomer supply and drive protocell growth, must have preceded those catalyzing templated polymerization. This conclusion is grounded in the assumption that protocells were autotrophic, synthesizing their own monomers, as this links catalysis of growth to increased ability to explore sequence space. On this robust path to genetic heredity, growth came first—setting the stage for the evolution of genetic information.

## Supporting information

**S1 Text. Model structure.** Detailed description of the model.
(DOCX)

**S1 Fig. Time-course of a model simulation.** (**a**) Evolutionary change in the rate of protocell divisions (per 50 time steps) for low ($p_c$ = 0.001, blue lines) and high ($p_c$ = 0.1, orange lines) baseline probability of copying. Thin lines show individual populations and thick lines the mean. The mean distribution of nucleotides at three time points (t = 1,000, 4,000, and 8,000 shown by the red dotted lines, for (**b–d**) low ($p_c$ = 0.001) and (**e** and **f**) high ($p_c$ = 0.1) baseline probability of copying. Each square in the heatmap shows the log count of RNA molecules. The data and scripts used to generate this figure are available in the GitHub repository archived on Zenodo (https://doi.org/10.5281/zenodo.18940155, folder Figure S1).
(DOCX)

**S2 Fig. Effect of changing probability of decay ($p_d$) on protocell division rate, composition and distributions of RNA in protocells.** (**a**) Changes in the moving sum of protocell division in the population per time step are shown for $p_d$ = 0.0001 (blue), $p_d$ = 0.001 (orange), and $p_d$ = 0.01 (yellow). (**b–d**) show the mean distribution of nucleotides across protocells at t = 8,000 (shown as a red dotted line in panel (**a**)) for (**b**) $p_d$ = 0.0001, (**c**) $p_d$ = 0.001, and (**d**) $p_d$ = 0.01. Each square in the heatmap shows the log count of a nucleotide molecule composed of a number of purines (its position in the x axis) and a number of pyrimidines (its position in the y axis). All other parameter values are in Table 1. The data and scripts used to generate this figure are available in the GitHub repository archived on Zenodo (https://doi.org/10.5281/zenodo.18940155, folder Figure S2).
(DOCX)

**S3 Fig. Impact of probability of translation ($p_t$) on protocell division rate, composition, and distributions of RNA in protocells.** (**a**) Changes in the moving sum of protocell division in the population per time step are shown for $p_t$ = 0.0001 (blue), $p_t$= 0.001(orange), and $p_t$= 0.01 (yellow). (**b–d**) show the mean distribution of nucleotides across protocells at $t$=8,000 (shown as a red dotted line in panel (**a**)) for (**b**) $p_t$ = 0.0001 (**c**) $p_t$= 0.001, and (**d**) $p_t$= 0.01. Each square in the heatmap shows the log count of a nucleotide molecule composed of a number of purines (its position in the x axis) and a number of pyrimidines (its position in the y axis). All other parameter values are in Table 1. The data and scripts used to generate this figure are available in the GitHub repository archived on Zenodo (https://doi.org/10.5281/zenodo.18940155, folder Figure S3). (DOCX)

**S4 Fig. Impact of varying the length of effective catalysts.** (**a**) Catalytic rate ($k_l$) as a function of peptide length (*l*) for increasing catalytic constants ($\gamma$ = 4 blue, $\gamma$ = 6 orange, $\gamma$ = 8 yellow, and $\gamma$ = 10 purple). (**b**) Evolutionary change in the rate of protocell divisions (per 50 time steps) for different values of $\gamma$. (**c–f**) show the mean Distribution of nucleotides across protocells at $t$ = 8000 for $\gamma$ = 4, (d) $\gamma$ = 6, (e) $\gamma$ = 8, and (f) $\gamma$ = 10. Each square in the heatmap shows the log count of RNA molecules composed of specific numbers of purines and pyrimidines. All other parameter values are as given in Table 1. The data and scripts used to generate this figure are available in the GitHub repository archived on Zenodo (https://doi.org/10.5281/zenodo.18940155, folder Figure S4). (DOCX)

**S5 Fig. Evolution of protocells with different catalytic optima for $CO_2$ fixation and templated polymerization.** Panel (**a**) shows the moving sum of protocell divisions over 50 time steps, and panels (**b** and **c**) show the average RNA compositions of protocells at time step 8,000. Catalytic rates ($k_l$ and $k_h$) depend on the optimal catalytic length ($\gamma$) and hydrophobicity ($\beta$) for $CO_2$ fixation ($\gamma_{fix}$, $\beta_{fix}$) and for templated polymerization ($\gamma_{pol}$, $\beta_{pol}$). In panel (b) (blue line in panel a), $\gamma_{fix}$ = 6, $\gamma_{pol}$ = 8, $\beta_{fix}$ = 0.8, and $\beta_{pol}$ = –0.2. In panel (c) (orange line in panel a) $\gamma_{fix}$= 6, $\gamma_{pol}$ = 8, $\beta_{fix}$ = 0.8, and $\beta_{pol}$ = –0.2. Each square in the heatmaps shows the log count of RNA molecules with a given number of purines and pyrimidines. All other parameters are listed in Table 1. The data and scripts used to generate this figure are available in the GitHub repository archived on Zenodo (https://doi.org/10.5281/zenodo.18940155, folder Figure S5). (DOCX)

**S6 Fig. The effect of varying the probability of random polymerization ($p_p$).** This is investigated when $CO_2$ fixation catalysts are easier to achieve at random ($\gamma_{fix}$= 6, $\gamma_{pol}$= 8, $\beta_{fix}$ = 0.2, and $\beta_{pol}$= –0.8). (**a**) Evolutionary change in the rate of protocell divisions (per 50 time steps) for $p_p$ = 0.01 (blue), $p_p$ = 0.04 (orange), $p_p$ = 0.08 (yellow), and $p_p$ = 0.12 (purple). The mean distribution of nucleotides across protocells at $t$ = 8,000 (red dotted line in panel (a)) is shown for (**b**) $p_p$ = 0.01, (**c**) $p_p$ = 0.04, (**d**) $p_p$ = 0.08, and (**e**) $p_p$ = 0.12. Each square in the heatmap shows the log count of RNA molecules composed of specific numbers of purines and pyrimidines. All other parameter values are in Table 1. The data and scripts used to generate this figure are available in the GitHub repository archived on Zenodo (https://doi.org/10.5281/zenodo.18940155, folder Figure S6). (DOCX)

## Acknowledgments

We thank Stuart Harrison and Aaron Halpern for discussions about the origin of heredity that were valuable background to the work presented here.

## Author contributions

**Conceptualization:** Raquel Nunes Palmeira, Marco Colnaghi, Andrew Pomiankowski, Nick Lane.

**Formal analysis:** Raquel Nunes Palmeira, Marco Colnaghi, Andrew Pomiankowski, Nick Lane.

**Funding acquisition:** Andrew Pomiankowski, Nick Lane.

**Investigation:** Raquel Nunes Palmeira, Marco Colnaghi, Andrew Pomiankowski, Nick Lane.

**Methodology:** Raquel Nunes Palmeira, Marco Colnaghi, Andrew Pomiankowski, Nick Lane.

**Writing – original draft:** Raquel Nunes Palmeira, Andrew Pomiankowski, Nick Lane.

**Writing – review & editing:** Raquel Nunes Palmeira, Marco Colnaghi, Andrew Pomiankowski, Nick Lane.

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
