## [Editor Report · Decision Letter 0]

12 Nov 2025

Dear Dr Pomiankowski,

Thank you for submitting your manuscript entitled "First growth, then information: the path to genetic heredity in protocells" for consideration as a Research Article by PLOS Biology.

Your manuscript has now been evaluated by the PLOS Biology editorial staff, and I'm writing to let you know that we would like to send your submission out for external peer review.

Once your full submission is complete, your paper will undergo a series of checks in preparation for peer review. After your manuscript has passed the checks it will be sent out for review. To provide the metadata for your submission, please Login to Editorial Manager (https://www.editorialmanager.com/pbiology) within two working days, i.e. by Nov 14 2025 11:59PM.

Kind regards,

Roli Roberts

Roland Roberts, PhD

Senior Editor

PLOS Biology

rroberts@plos.org

---

## [Decision Letter · Decision Letter 1]

14 Jan 2026

Dear Dr Pomiankowski,

Thank you for your patience while your manuscript "First growth, then information: the path to genetic heredity in protocells" was peer-reviewed at PLOS Biology. It has now been evaluated by the PLOS Biology editors, an Academic Editor with relevant expertise, and by two independent reviewers. Please accept my apologies for the delay incurred due to our two-week closure over the holiday period.

In light of the reviews, which you will find at the end of this email, we would like to invite you to revise the work to thoroughly address the reviewers' reports.

You'll see that reviewer #1 is broadly positive, but wonders if the study’s apparent intuitiveness (which might be seen as triviality?) might limit its appeal; as a result, s/he suggests re-framing it by presenting the key dilemma upfront. Reviewer #2 says enjoyed reading the paper, but raises several significant concerns about the modelling that will need to be addressed; s/he also wants more clarity about the simulations in the main paper and asks about code availability (which is mandatory under PLOS policy).

Given the extent of revision needed, we cannot make a decision about publication until we have seen the revised manuscript and your response to the reviewers' comments. Your revised manuscript is likely to be sent for further evaluation by all or a subset of the reviewers.

**IMPORTANT - SUBMITTING YOUR REVISION**

*Re-submission Checklist*

*Published Peer Review*

*PLOS Data Policy*

*Blot and Gel Data Policy*

Sincerely,

Roli Roberts

Roland Roberts, PhD

Senior Editor

PLOS Biology

rroberts@plos.org

REVIEWERS' COMMENTS:

Reviewer #1:

"First growth, then information: the path to genetic heredity in protocells" by Palmeira et al presents an exceptionally simple model for the evolution of metabolism, which is equated to protocell growth, and evolution of templated synthesis (here, equivalent to heredity) at the stage of(pre)life evolution.

The simplicity of the model is quite striking: hydrophobic peptides are assumed to be responsible growth (CO2 fixation) whereas hydrophilic ones are assigned to templated synthesis. The outcome of the simulations the authors perform under this model is quite obvious: for the protocells to reach a high division rate, the cost of growth has to be lower than the cost of templated synthesis. Hence 'First growth, then information'. This conclusion holds for autotrophic protocells, that is, in the absence of influx of monomers from the environment. In the heterotrophic version of the model, evolution of metabolism can be uncoupled from the evolution of information transmission unless the external supply of monomers is rate-liming.

This work is quite easy to read and understand but somewhat difficult to evaluate. On the one hand, the results are so intuitive that they potentially might be considered trivial. Indeed, how can replication and translation be possibly sustained if there is not steady, sufficient supply of monomers and energy...

However, exactly because of its simplicity, this work might become impactful in the origin of life field where 'metabolism first vs replication first' is still viewed as a serious conundrum. Hence my suggestion to the authors is to present this dilemma explicitly from the start, to set up the context for the modeling approach. This way, the work may be considered as a step towards the solution of an important problem in the origin of life research, and as such could be of interest to many.

Reviewer #2:

In "First growth, then information: the path to genetic heredity in protocells" the authors develop and use a mathematical model to understand how RNA polymers in protocells could evolve longer coding sequences. Motivating this work is Eigen's paradox in which error-prone copying of sequences needs error-proofing sequences, but these are so long that they themselves would also need error-proofing sequences. While other models have addressed issues of preserving sequence information the authors raise questions concerning the purposes/functions of early sequences. The authors address these questions by considering a scenario in which growth and information preservation are coupled. Specifically, their model considers two essential processes affecting polymer production: the fidelity of sequence replication (information) and the production of monomers for sequence construction (growth). They study scenarios in which growth or information processes are easier to evolve by random chance. Importantly selection acts at the level of the protocell instead of just the sequences. Through their analyses they find that when sequences promoting growth are easier to evolve it leads to both growth and information-preserving sequences. Instead when sequences promoting information fidelity are easier to evolve, it stymies the population from evolving longer sequences. Overall, I enjoyed this paper and think its results in terms of concept and significance are suitable for publication. I have some questions concerning the modeling that should be addressed.

In terms of the modeling, many of the assumptions regarding the formulation of the model were well-justified. Certainly one could have done a more explicit model that took into account the actual sequences of the polymers, but that would likely introduce many additional parameters and complexity without fundamentally changing the main result. That said, I did have issues/questions with the modeling that I would like the authors to address.

First, in equation 3 in the Supplement there is an equation describing the average change in the number of polymers of a certain length and hydrophobocity. In this equation, binding occurs based on a certain probability and the product of three terms: 1. the number of a type of polymer and 2,3. the fractions of two types of polymers in the current population. This formulation seems a bit unusual. For one, scaling the terms by the current population of nucleotides is a different normalization than typically used. In mass-action models the normalization is often done by volume so that the variables are concentrations. If the volume is constant then this would mean that the binding probability term would just have different units. Instead, the authors scale by the population size, which affects the kinetics. It is effectively sampling from the existing polymer pool (with frequency-dependence) rather than using mass-action kinetics in a fixed volume.

It is also not clear to me why the N_{l-1,i-1} term appears twice. To put it more clearly, suppose there were two types of molecules that might interact to give rise to a third, e.g. X+Y --p Z. A differential equations model of this might look something like dZ/dt= p[X][Y]. Yet, adopting the approach in this paper, the model would look something like dZ/dt=p[X] [X]/[X+Y+Z] [Y]/[X+Y+Z]. Perhaps the authors are making some assumptions that led to this model choice but from reading the text it is not obvious to me what they are.

Second, disregarding incomplete RNA copies removes a potentially important source of shorter sequences that could possibly outcompete longer molecules. Including such sequence fragments could alter the balance between sequence construction and decay and thus change the steady state distribution.

Similarly, there is an assumption that all polymers have a probability "pd" of losing a monomer, which is weighted by length. Yet if a longer polymer has a higher chance of losing a monomer then it seems losing a middle monomer would generate two smaller polymer sequences instead of a monomer and polymer. I believe this could be fixed by simply assuming that monomer loss can only occur at the ends, though I don't know if there is any empirical motivation for this assumption.

Finally, most of the details of the model were in the Supplement. While I understand why this choice was made, it was challenging to understand how exactly the simulation worked. Perhaps there is computer code in some public repository? Reading the main text and the supplement, I think I have some sense of what is happening but I am unsure if I could precisely replicate their model. More clarification is needed, especially in the main text. Such clarification is important because the results of the paper rely on these simulations. The authors have a schematic for the main processes in the protocell in Figure 1, but I think some more information would help clarify the structure of the model. For example, it would be useful to indicate that there is a population of protocells undergoing evolution via a Moran process. Also, it would be useful to depict how cell reproduction occurs, i.e. when the fatty acid population exceeds some threshold. Last, it would be helpful to indicate the various sources and sinks for monomers---especially because these are manipulated in subsequent figures.

Minor points:

The figure captions could benefit from some sentences describing the central result that should be observed. When reading this paper, I had to keep alternating between figure and sections in the Results to understand what I should be noting.

In S1.2 I think there is a typo in "with m_{max}ials "

---

## [Decision Letter · Decision Letter 2]

3 Mar 2026

Dear Dr Pomiankowski,

Thank you for your patience while we considered your revised manuscript "First growth, then information: the path to genetic heredity in protocells" for publication as a Research Article at PLOS Biology. Please note that I am currently handling your manuscript since my colleague Roland Roberts is out of the office this week. I am sorry for the delays that you have experienced during this round of the peer review process. This revised version of your manuscript has been evaluated by the PLOS Biology editors, the Academic Editor and Reviewer #2. Please note that Reviewer #2 did not provide any specific comments but recommended that we accept your manuscript for publication.

Based on the review from Reviewer #2, I am pleased to say that we are likely to accept this manuscript for publication, provided you satisfactorily address the following data and other policy-related requests that I have provided below (A-F):

(A) We routinely suggest changes to titles to ensure maximum accessibility for a broad, non-specialist readership. In this case, we would suggest the following edit to the title, as follows. Please ensure you change both the manuscript file and the online submission system, as they need to match for final acceptance:

“Evolution of genetic heredity is driven by metabolic constraints in early protocells”

(B) Please note that we cannot accept sole deposition of code in GitHub, as this could be changed after publication. However, you can archive this version of your publicly available GitHub code to Zenodo. Once you do this, it will generate a DOI number, which you will need to provide in the Data Accessibility Statement (you are welcome to also provide the GitHub access information). See the process for doing this here: https://docs.github.com/en/repositories/archiving-a-github-repository/referencing-and-citing-content

(C) In the Github deposition, we note that the readme lists testMatlabFunction.m as an entry point, but we could not find a file with this name in the repository?

(D) Thank you for providing the scripts to generate the figures in the manuscript in the Github deposition. This looks good but I would be grateful if you could ensure that scripts are provided for the supplementary figures as well. The figure panels that we will need underlying data for are as follows:

Figure 2A-C, 3A-C, 4A-B, 5A-E, S1B-D, S2A-G, S3A-D, S4A-D, S5A-F, S6A-C, S7A-E

(E) Please also ensure that each of the relevant figure legends in your manuscript include information on *WHERE THE UNDERLYING DATA CAN BE FOUND*, and ensure your supplemental data file/s has a legend.

(F) Please ensure that your Data Statement in the submission system accurately describes where your data can be found and is in final format, as it will be published as written there.

We expect to receive your revised manuscript within two weeks.

*Published Peer Review History*

*Press*

Best regards,

Richard

Richard Hodge, PhD

rhodge@plos.org

On behalf of:

Roland Roberts, PhD

rroberts@plos.org

---

## [Editor Report · Decision Letter 3]

18 Mar 2026

Dear Pom,

Thank you for the submission of your revised Research Article "Selection for growth drives the emergence of genetic heredity in protocells" for publication in PLOS Biology. On behalf of my colleagues and the Academic Editor, Claudia Bank, I'm pleased to say that we can in principle accept your manuscript for publication, provided you address any remaining formatting and reporting issues. These will be detailed in an email you should receive within 2-3 business days from our colleagues in the journal operations team; no action is required from you until then. Please note that we will not be able to formally accept your manuscript and schedule it for publication until you have completed any requested changes.

Sincerely,

Roli

Senior Editor

PLOS Biology

rroberts@plos.org